# Microorganism Contribution to Mass-Reared Edible Insects: Opportunities and Challenges

**DOI:** 10.3390/insects15080611

**Published:** 2024-08-13

**Authors:** Joachim Carpentier, Linda Abenaim, Hugo Luttenschlager, Kenza Dessauvages, Yangyang Liu, Prince Samoah, Frédéric Francis, Rudy Caparros Megido

**Affiliations:** 1Functional and Evolutionary Entomology, Gembloux Agro-Bio Tech, University of Liège, Passage Des Déportés 2, 5030 Gembloux, Belgium; joachim.carpentier@uliege.be (J.C.); hluttenschlager@uliege.be (H.L.); kenza.dessauvages@uliege.be (K.D.); yangyang.liu@doct.uliege.be (Y.L.); samoahprince@gmail.com (P.S.); frederic.francis@uliege.be (F.F.); 2Department of Agriculture, Food and Environment, University of Pisa, Via del Borghetto 80, 56124 Pisa, Italy; linda.abenaim@phd.unipi.it; 3Institute of Feed Research, Chinese Academy of Agricultural Sciences (CAAS), Haidian District, Beijing 100193, China

**Keywords:** edible insects, midgut, intestinal microbiota, microbial community, valorization of by-products, detoxification

## Abstract

**Simple Summary:**

Interest in large-scale rearing edible insects such as beetles, crickets, and flies has increased significantly in recent years. These insects are now used for various purposes: as food and feed, managing organic and plastic waste, detoxifying environments, producing biofuels, and even in cosmetics and pharmaceuticals. These applications consist of feeding insects with waste materials that are not widely used, transforming them into valuable products like food, feed, and fertilizer. The insect’s digestive system is therefore the keystone of these developing processes. Digestion is partly carried out by the insect itself and partly by gut-associated microorganisms. Their respective roles remain a needed research area, and it is now clear that the community of microorganisms can adapt, enhance, and extend the insect’s ability to digest and detoxify their feed. Despite this, these species are surprisingly autonomous, with no mandatory association with microorganisms required for digestion. On the contrary, microbiota largely differ for the same species, and are mostly shaped by the host’s environment and diet. This natural flexibility offers the prospect of targeting and developing novel associations between insects and microorganisms to create mass-reared strains tailored to manage specific by-products and industrial applications.

**Abstract:**

The interest in edible insects’ mass rearing has grown considerably in recent years, thereby highlighting the challenges of domesticating new animal species. Insects are being considered for use in the management of organic by-products from the agro-industry, synthetic by-products from the plastics industry including particular detoxification processes. The processes depend on the insect’s digestive system which is based on two components: an enzymatic intrinsic cargo to the insect species and another extrinsic cargo provided by the microbial community colonizing—associated with the insect host. Advances have been made in the identification of the origin of the digestive functions observed in the midgut. It is now evident that the community of microorganisms can adapt, improve, and extend the insect’s ability to digest and detoxify its food. Nevertheless, edible insect species such as *Hermetia illucens* and *Tenebrio molitor* are surprisingly autonomous, and no obligatory symbiosis with a microorganism has yet been uncovered for digestion. Conversely, the intestinal microbiota of a given species can take on different forms, which are largely influenced by the host’s environment and diet. This flexibility offers the potential for the development of novel associations between insects and microorganisms, which could result in the creation of synergies that would optimize or expand value chains for agro-industrial by-products, as well as for contaminants.

## 1. Introduction

The interest in mass-reared insect research for human and animal consumption has exponentially increased in the last five years. In particular, the search of the “edible insect” keyword in the Scopus database (https://www.elsevier.com/solutions/scopus (accessed on 4 August 2024) shows 3377 research papers from 1973 to January 2024 and the number of publications has been growing since 2020 (280 publications in 2021, 315 in 2022 and 533 in 2023). This stronger attention is due to several sustainable advantages related to qualities of edible insect rearing such as ensuring the increase in global population with fewer resources to satisfy the increasing demand for food, the increase in demand for animal protein but also less impacts on climate changes and environmental degradation [1]. Edible insects should be an interesting source of protein, amino acids, fatty acids, minerals, vitamins and other nutrients, including bioactive compounds for animals and humans [1,2,3,4,5,6,7,8]. Besides, they seem to be more sustainable and environmentally friendly than other livestock animals because their mass rearing produces less greenhouse gas emissions, requires less use of water, soil and feed and shows a lower feed conversion rate than intensive farming [1,9,10,11]. Furthermore, edible insects have a high reproduction rate and a wide geographical distribution which could contribute to the choice of these as feed and food ingredients [12]. According to statistics, there are around 2000 species of edible insects, out of a total of over one million insect species, principally presented in Africa, Asia, and the Americas [13,14]. Ongoing research will likely only further increase the number of insects suitable for consumption [13,15]. The primary species reared for feed and food production on a semi-industrial or industrial scale are Coleoptera (*Tenebrio molitor* L. 1758 (Tenebrionidae), *Alphitobius diaperinus* Panzer 1797 (Tenebrionidae) and *Zophobas* (*Zophobas*) *atratus* (Fabricius 1775)), Orthoptera (*Acheta domesticus* L. 1758 (Gryllidae), *Gryllodes sigillatus* (Walker 1869) (Gryllidae), *Gryllus bimaculatus* De Geer 1773 (Gryllidae) and *Locusta migratoria* (L. 1758) (Acrididae)), Lepidoptera (*Galleria mellonella* (L. 1758) (Pyralidae) and *Bombyx mori* (L 1758) (Bombycidae)) and some Diptera (*Hermetia illucens* L. 1758 (Stratiomyidae) and *Musca domestica* L. 1758 (Muscidae)) [16,17]. Edible insects have been studied in many aspects, such as nutritional values [18,19], sustainability [20,21], human and animal health [17,22], food safety [15,23], rearing [24,25] and intestinal microbiota [26,27].

### 1.1. Insect Microbiota

Interestingly, publications demonstrating microbial associations with edible insects have steadily increased over the last five years with a steeper increase in the past three years [28]. This is in line with the new approach in animal domestication developed which no longer considers animals alone but as an ecological functional unit formed by a multicellular organism and symbiotic microorganisms called an “ecosystem-holobiont” [29,30]. The entomological diversity of digestive systems provides a wide range of habitat opportunities for microorganisms with abundant food, low oxygen, and varying degrees of acidity [31,32,33,34]. In insects, the microbiota is involved in host nutrition, immunity, and detoxification and the insect effectiveness is directly related to the state of their microbial communities [29,35,36]. In contrast to mammals, co-species events with microorganisms in insects, except for social species such as Hymenoptera and termites, primarily involve maternally transmitted endosymbiosis relationships [37,38]. These inherited bacteria are found in tissues (i.e., bacteriomes) and cells (i.e., bacteriocytes) [38]. Insects lacking social structures exhibit limited individual and intergenerational interactions. Extracellular transmission within a host lineage can occur horizontally, such as through coprophagy, or vertically during oviposition Figure 1 [39,40]. This transmission route deficit leads to intraspecific variability in the communities of microorganisms colonizing the insect digestive system [34,41]. Microorganism acquisition occurs over the entire developmental lifespan of the insect through environment and diet. At each larval stage change, molting cause an abrupt loss of extracellular microbiota by shedding the exoskeleton from the anterior and posterior parts of the digestive tract Figure 1 [34]. However, the host can reacquire its microbiota by consuming its exoskeleton or through its diet and environment [42]. The loss caused by metamorphosis is almost total in holometabolous insects as it may facilitate a shift in habitat and diet allowing a new community to establish itself in the gut [43,44].

Farmed insects seem to be independent of microorganisms for their nutrition and reproduction since they are neither sap feeders nor social insects like termites [34,45]. Therefore, the “ecosystem-holobiont” should be adjustable under rearing conditions, and the acquired microbiota could be conditioned to enhance the digestive processes and immunity of the rearing population. Martino et al. [46] suggest that the prerequisite step for a new facultative symbiosis is the adaptation of the bacterial strain to the host diet. This modulation is constrained by the characteristics of the insect as its developmental mode, dietary requirements, and digestive system [34,41,44]. Previous studies have identified microbial biodiversity in the gut of edible insects such as *H. illucens* or *Bombyx mori* and postulated that the microbiome diversity results from the substrate and the larval development stage [46,47,48,49,50]. In most reviewed insect species (proposed as edible insects in Van Huis [17]) or other insect species from the same taxonomic family, phyla of bacteria such as Firmicutes, Proteobacteria, and Bacteroidetes are dominant [51,52,53,54]. The comprehension of insect species bioconversion or detoxification capacity could be linked with the microbial community of the population but, above all, must be linked with the function covered by this community [34]. The microbial functions during insect’s digestion could be the basis of the observed substrate-dependent microbial community structure variations, as the bacteria more adapted to digest a specific substrate will have a fitness advantage in the gastrointestinal tract (GIT) over less-adapted bacteria [55]. Microbiotas have an important role in facilitating the digestion of the nutrients present in the insect diet, making them more easily digestible for the insect. However, microorganisms not only contribute to the nutritional needs of insects but also play a role in protecting them against natural enemies and detoxifying food compounds including contaminants. Moreover, these insects are also considered bioreactors with broad biotechnological potential for the valorization of recalcitrant or contaminated by-products [36,56]. Specifically, this review wants to investigate the contribution of intestinal microbiota on the edible insect’s digestion process of natural and synthetic molecules such as carbohydrates, proteins, lipids, vitamins and minerals, and contaminants.

### 1.2. Insect Digestive System

The insect digestive system is responsible for all steps of food processing: ingestion, digestion and absorption as well as elimination of undigested parts of food. These steps occur along the gut comprising three principal areas: the stomodaeum or foregut, the mesenteron or midgut, and the proctodaeum or hindgut Figure 1.

The foregut, as well as the hindgut, is of ectodermic origin and is characterized by a cuticle that covers the cell layer (epithelium). The foregut is the anterior part of the gut and is open in the oral cavity. It is constituted by the pharynx, oesophagus, crop (food storage area) and proventriculus (a grinding area not always present). In the foregut, food is ingested, stored, minced and conveyed to the midgut. The latter is the middle part of the digestive system, commonly called the ventriculus, and is where digestion and absorption occur. The flow of food between the foregut and midgut is regulated by the cardiac valve at the end of which, in the high midgut, gastric caeca full of symbiotic fauna are present [57]. The midgut, as well as caeca walls, do not have a cuticle but, towards the lumen, present only an epithelium with a rhabdorium (brush border) which facilitates the secretion of enzymes and the absorption of nutrients by the cells of the epithelium [58]. The insects are characterized by a full complement of ordinary digestive enzymes which are synthesized and secreted by the midgut cells, and the amount of each enzyme depends on the secretory rate. The digestive enzymes change in response to the feeding habits and adaptations of the taxonomic group to a particular diet [59]. However, other enzymes produced by microbial communities of caeca, which are in a mutualistic association with the gut of insects, contribute critically to the digestion of food components in addition to performing a fundamental role in the development, morphogenesis, immunity and behavior of insects [34,56,60]. In the midgut, the delicate rhabdorium is isolated and protected from the abrasion of a coarse bolus by the peritrophic membrane, which is a dialyzing, dense netting of thin cuticular fibrillae produced temporarily and eliminated with feces [61]. In addition, the peritrophic membrane improves digestion by moving digestive enzymes and still partially digested material in counter-current flows [62]. Subsequently, the digestive enzymes and nutrients are moved by the counter-current fluxes to the excretory organs called the Malpighian tubules. They are placed at the end of the midgut, at the beginning of the hindgut, or the junction of the two, and are responsible for the production of excreta and other alternative functions [63]. The hindgut is the last part of the digestive system in which water and salts are recovered from the feces. Three sections of the hindgut (ileum, colon and rectum), each separated from the other by a valvula, are visible [64]. As soon as the bolus from the midgut passes into the hindgut, digestion ends, and fecal formation occurs [59,62,64,65].

### 1.3. Edible Insects’ Intestine

Many studies summarized the principal differences in the gut structures between insect taxa. However, these are often related only to the shape of the different intestinal tracts more than to their functions. Actually, except for particular cases (such as the filter chamber of the Homoptera, and the interrupted gut of the Diaspididae), the functions of the different tracts are slightly unchanged regardless of the shape. The microbial communities in insects are primarily concentrated in the midgut and hindgut. The midgut hosts microorganisms that contribute to the digestion and absorption of nutrients, even though the environment can be unfavorable for bacterial proliferation due to the presence of digestive enzymes, loss of the peritrophic membrane and its associated bacteria, and variable pH levels [34]. In contrast, the hindgut, as the final part of the intestine rich in waste products, promotes the growth and diversity of the intestinal microbiota, hosting a particularly diverse and abundant community of microorganisms [66]. The composition of intestinal microorganisms is influenced by several factors, including the insect species, developmental stage, and biochemical conditions [66]. Generally, the gut microbiota of insects includes archaea, bacteria, fungi and protists. Protists are primarily found in the intestines of termites, fungi and archaea in the intestines of wood-feeding insects, and bacteria mainly in omnivorous ones [34,66]. Therefore, diet is a significant factor affecting the diversity of gut microbiota in various insect species. A large variety of bacterial phyla are present in the intestines of insects, including Actinobacteria, Actinomycetes, Alphaproteobacteria, Bacteroidetes, Betaproteobacteria, Firmicutes (with species such as *Lactobacillus* spp. and *Bacillus* spp.), Gammaproteobacteria, Spirochetes and Verrucomicrobia, among others [34].

As edible insect species mainly belong to Orthoptera, Coleoptera, Diptera and Lepidoptera orders, the gut tracts present limited differences in function and shape. In Orthoptera of the Caelifera sub-order as *L. migratoria*, mainly feeding on grass, the cellulase enzyme allows the degradation of cellulose in the midgut but carbohydrates and proteins are digested principally in the crop and caecal lumina, respectively, by the digestive enzymes driven by antiperistalsis from the midgut, while the final digestion of proteins occurs in the caecal and ventricular cells [62,67]. Regarding the morphology, in *L. migratoria*, the proventriculus is not a distinct region but rather a continuum of the crop. The midgut is characterized by six gastric caeca. Meanwhile, in the hindgut, the ileum lies along the axis of the body, and the colon assumes an S shape form [68]. Generally, in the Ensifera sub-order, the gut does not present any peculiar differences compared to the basic structure [69]. This is also confirmed by the previous description of the anatomical features of the gut of *A. domesticus* by Kirby et al. [70]. Also, in Ensifera, omnivorous or predators of many insects, the digestive enzymes are conveyed forward, from the anterior midgut to the crop, by peristalsis. Many Coleoptera have a reduced or absent crop, and the protein digestion occurs only in the midgut cells. In Coleoptera, enzyme recycling occurs at the end, while absorption starts at the beginning of the midgut, respectively [62]. In the gut of *T. molitor*, the peritrophic membrane is present along all the midgut, and the entire digestive process seems to occur inside the endoperitrophic space at the level of the anterior midgut, except for the digestion of protein which occurs partly inside the peritrophic membrane and partly over the cell surface of the posterior midgut [59,71]. The relevant difference with other order regards water recovery from the feces. For that, some coleopterans species have a cryptonephric structure which is an arrangement of Malpighian tubules that, for the more significant part of their length, are as usual, free in the haemocoel and only the ends of the tubules are bound to the wall of the rectum by a perinephric membrane. The high concentration of ions, present in the Malpighian tubules, draws water from the rectum by osmosis. This represents a highly efficient mechanism for water recovery in insects such as *Tenebrio* spp. that feed on very dry materials (cereals and foodstuff products) [72,73,74]. Even *A. diaperinus* presents a cryptonephric system. In particular, the digestive system of *A. diaperinus* consists of a straight tube with attached six Malphighian tubules, which are equally spaced around the hindgut in larvae while in adults, are attached vertically and dorsally [75]. Unlike *T. molitor*, the *A. diaperinus* midgut epithelium lacks regenerative crypts. This lack provides evidence of the scavenging and omnivorous habits of the species, which usually eats small, soft remains of organic materials which differentiates the species from stored product pests which usually eat coarse pieces of dry grain. Apart from this difference, the *A. diaperinus* gut does not differ significantly from the baseline system [75]. Lepidoptera’s gut shows no peculiarities apart from the absence of caeca, and all digestive enzymes are therefore produced by the cells of the midgut epithelium [62,65,76]. Cryptonephric system should also be found in the hindgut region of *B. mori* larvae which is responsible for electrolyte balance and formation and elimination of faecal pellets [77].

Diptera developed an evolved digestive system to support fast development [62]. As in all Diptera, in which the remodelling of the larval midgut is a key process that occurs during metamorphosis, the midgut shows significant changes along the preimaginal instars phases. The anterior larval midgut is characterized by an infolded epithelium that is not present in the middle midgut, which is short and narrow. In larvae, gastric caeca are lacking, and the posterior midgut is characterized by constriction [78]. The adult midgut presents the anterior and posterior parts as thin and tubular structures, while the middle midgut appears more extensive than the other two. The peritrophic membrane is not present and the lumen content is in direct contact with the brush border of the epithelium cells. The lack of peritrophic membrane has been related to the feeding habits of the adults, which feed liquids, and thus the midgut brush does not require any protection. In *H. illucens* adults, the digestion of carbohydrates and lipids occurs principally in the anterior midgut, while protein digestion occurs in the posterior tract [31]. Diptera from the Brachycera sub-order (as *H. illucens*) are often organic matter degraders and are adapted to diets contaminated by numerous bacteria with a digestive system adapted to live in adverse environments. However, during the digestion process, bacteria are eliminated. First, in the anterior midgut, in which digestion of dietary carbohydrates such as starch takes place, there is a decrease in the presence of bacteria directly related to the decreasing availability of growth substrate. As soon as the food goes through the middle midgut, bacteria are killed by the low pH and by the action of some enzymes such as lysozyme. Digestion of proteins, polysaccharides and lipids in *H. illucens* larvae takes place in the posterior midgut and the counter-current flow of enzymes carries the materials back to the middle midgut where nutrient absorption takes place [62,65]. The anterior and middle midgut of all Brachycera (thus also in *H. illucens*), is acidified, which contributes to killing the pathogenic microorganisms ingested with diet, while the posterior midgut content is alkaline.

## 2. Role of Microorganism in the Digestive Process of Natural and Synthetic Organic

### 2.1. Carbohydrates

Carbohydrates are organic molecules that play a vital role in the metabolic cycles of organisms. They can be unitary (ose), polymerized (oligosides and polyosides) or combined with other types of molecules to form heterosides. Their degree of polymerization determines their role. The oses are considered direct sources of energy for the organisms, while polyosides, made up of multiple units, may be either energy storage (e.g., starch and glycogen in plants and animals, respectively) or structural components (e.g., cellulose and lignin forming plant cell walls). The level of complexity of the latter differs according to plant species and part of the plant. Tissues originating from primary plant growth typically exhibit lower cellulose composition, whereas those from secondary growth tend to develop into woody structures. The degradation of plant walls in non-woody parts is ensured by endogenous enzymes in basal insect orders and arthropod groups closely related to insects. These enzymatic capacities reflect the ancestral autonomy of insects to independently digest these plant parts without relying on microorganisms [79]. In contrast, these insect-digestible polyosides are trapped in tough walls in woody plants, which are much less digestible [80]. In this case, the pre-degradation of the wood by a fungus facilitates access to the sugars of interest in wood-eating insects [34]. In mass-reared insects, carbohydrates often represent the bulk of the macronutrients making up their diet (spent grain, vegetable trimmings, cakes, straw, and others). Most of the by-products used in diets consist of plant components from which a desired product has been extracted. Therefore, they comprise a non-extractable fraction (carbohydrate, protein or lipid) from the original product, as well as the structural and storage of the plant [81]. As far as absorption is concerned, only oses and diosides can be directly assimilated by the insect’s gut [82,83,84]. Consequently, reserve and structural polyosides must undergo enzymatic lysis in the digestive system before they can be absorbed and metabolized by the organism [85]. The number and organization of genes encoding enzymes linked to metabolic pathways differ between insect species, reflecting the specialization of their respective metabolism. Insects are capable of producing alpha-amylases, alpha/beta-glucosidases and endo/ex-beta-1,4-glucanases [82]. This specialization seems to exhibit a greater emphasis on carbohydrate metabolism than on lipids [86]. For amylase, for example, several enzymes with the same function can be produced to cope with anti-nutritional factors and variable pH conditions in the gut [87]. In contrast, complex structural compounds such as cellulose, lignocellulose, and lignin pose a challenge to insects, except for xylophagous species such as termites. Initially, it was widely believed that their capacity to digest woody compounds relied solely on obligatory symbiosis with microorganisms, until the discovery of endogenous cellulases [88]. These cellulases are a complex of three enzymes: endo-glucanase, exoglucanase and beta-glucosidase [89]. Carbohydrate digestion is shared between the individual and its microbial community; however, symbiosis with the latter seems to remain obligatory in these social xylophagous insects.

The gut microorganisms play a multi-tasking role in insect carbohydrate digestion (Table 1). Firstly, polyosides can be lysed into oses that can be assimilated. The importance of this role seems to differ from that found in mammals. Wong et al. [90] revealed that the microbiota does not enhance carbohydrate availability in Drosophila sp. and several modes of action have been identified. For example, in anaerobic bacteria of the Clostridia class, prokaryotic cells produce cellulosomes, extracellular organelles capable of digesting cellulose and used by the bacteria to adhere directly to the cellulose substrate. Some of these cellulosomes are not anchored to the cell wall and act against cellulose independently of the bacterial cell [89]. Secondly, bacteria can also assimilate ingested carbohydrates reducing the insect’s consumption [91]. Then, oses and diosides can be metabolized through different fermentation pathways and produce ethanol, volatile fatty acids and CO_2_ depending on the bacterial species. Among the bacteria capable of oxidizing ethanol to acetic acid, the genera *Acetobacter*, *Gluconacetobacter* and *Gluconobacter* are naturally present in the digestive systems of insects consuming plant substrates and sugary feed [92]. The abundance and diversity of the cellulolytic bacterial community are positively correlated with the larval stage and carbohydrate heterogeneity of the insect diet [82,93,94]. For example, a diet rich in carbohydrates increases *Bacteroidota* abundance in *H. illucens* larvae [31]. Among this taxon, Bruno et al. [31] suggested a high involvement of the bacterial genera *Sphingobacterium* and *Dysgonomonas* in the degradation of complex polysaccharides based on their abundance in the posterior midgut of individuals fed a carbohydrate-rich diet. In *H. illucens*, aerobic isolation of gut bacteria revealed the presence of amylase- and cellulase-producing *Bacillus amyloliquefaciens* and *Bacillus stratosphericus*. However, metagenomic analysis showed a very low proportion of these species in the total gut bacterial community [48]. Indeed, Jeon et al. [48] isolated a *Bacillus* strain from the gut of *H. illucens* and observed its cellulolytic activity under aerobic conditions. If the microorganism was well present in the intestine, it cannot be concluded that it exhibited any significant cellulolytic activity within it. However, its dispersal through the substrate by maggots could be useful. Concerning the digestion of oses, *H. illucens* proved to be more efficient than conventional bioreactor-based microbial digestion of xylose-rich rice straw residues [81].

In *A. domesticus*, gut microorganisms are also capable of metabolizing soluble polysaccharides, which are composed of arabinose, xylose, mannose, galactose and glucose [95,96]. In 1981, Ulrich et al. [96] isolated two strains of *Fusobacterium* from the hindgut of *A. domesticus*. Interestingly, the glycolytic activities of the two strains proved to be complementary: one capable of metabolizing most oses and diosides, and the other capable of metabolizing mannitol and glycerol. For the degradation of these nutrients, microorganisms seem to favor acetate-, propionate- and butyrate-producing pathways in descending order [97]. The use of these metabolic pathways seems to be negatively correlated with an increase in protein levels in the *A. domesticus* diet, to the detriment of carbohydrate levels, since a decrease in the production of short-chain fatty acids and CO_2_ is observed [97].

In Coleoptera, the involvement of microorganisms in carbohydrates digestion seems less studied than in Diptera and Orthoptera. Nevertheless, the complete independence in polysaccharide digestion has been demonstrated in *T. molitor* [98,99]. In contrast, when heterosides such as salicyne are present in the diet, the microbial community remains essential for their detoxification [98]. Among other edible Coleoptera species, pests of palm trees have been more extensively studied to develop pest management methods. Recently, the gut microbiota of *Oryctes rhinoceros* (L. 1758) has shown significant involvement in the digestion of ligneous substrates [100]. In particular, the genera *Acetovibrio*, *Christensenella*, *Clostridium*, and *Ruminiclostridium* have been identified as cellulase producers [100]. Regarding *Rhynchophorus ferrugineus* (Olivier 1790), cellulolytic activities have been detected in the genera *Bacillus*, *Kocuria*, *Serratia*, and *Shigella* [101].

Concerning other mass-reared insect species, the glycolysis functions of the bacterial communities of *B. mori* and *Periplaneta americana* (L., 1758) are performed mainly by Gram+ bacteria, whose roles are relatively clear. In *B. mori*, they can degrade various types of polysaccharides from mulberry leaves, through the production of cellulolytic, xylanolytic, pectinolytic, and amylolytic enzymes. A similar observation was made in *P. americana*, since the presence of cellulose and sugarcane bagasse in its diet enhances the colonization of the digestive system by *Bacillota*, supporting their involvement in the degradation processes of these compounds [102]. Gram+ bacteria are the main producers of these four types of enzymes, but bacterial species are specialized in the degradation of one or two types of polyisodes. In contrast, *Bacillus circulans*, the only Gram+ bacterium involved in all glycolytic processes in *B. mori*, produces all four types of enzymes and shows the highest starch degradation activities [93].

Lastly, microorganisms other than bacteria are involved in the degradation of carbohydrates in insects. In termites, the degradation of cellulose and hemicellulose is shared between bacteria and flagellate protists [56]. This is also the case for *P. americana*, where the increase in carbohydrolase activity induced by a cellulose-rich diet is directly attributed to the increase in the protozoan *Nyctotherus ovalis* Leidy, 1850 colonizing the hindgut [103].

**Table 1 insects-15-00611-t001:** Gut bacteria of mass-reared insect species and their identified function related to carbohydrates digestion.

Insect Species	Gut Bacteria	Identified Function Related to Carbohydrates Digestion	Reference
*Acheta domesticus*	*Fusobacterium* sp.	Oses, diosides, mannitol and glycerol metabolization	[96]
*Bombyx mori*	*Bacillus circulans*	Amylolytic, cellulolytic, pectinolytic and xylanolytic activities	[93]
*Hermetia illucens*	*Dysgonomonas* sp.	Starch degradation	[31]
	*Sphingobacterium* sp.		
	*Bacillus amyloliquefaciens*	Amylase and cellulase pro-duction	[48]
	*Bacillus stratosphericus*		
*Oryctes rhinoceros*	*Acetovibrio* sp.	Cellulase production	[100]
	*Christensenella* sp.		
	*Clostridium* sp.		
	*Ruminiclostridium* sp.		
*Rhynchophorus ferrugi-neus*	*Bacillus* sp.	Cellulolytic activities	[101]
	*Kocuria* sp.		
	*Serratia* sp.		
	*Shigella* sp.		

### 2.2. Protein

Proteins are a very abundant nutrient that play an important role in building tissues and muscles, and they also have various essential roles in the circulatory system through oxygen transport and the endocrine system via hormone synthesis [104]. Insects generally need to consume an abundant quantity of protein during their growth stage in order to synthesize their exoskeleton and carry out their molt [105]. Insect exoskeletons are mainly made up of polysaccharides, but a high protein content gives them strength, support and flexibility [106]. Amino acids are the primary building blocks for the synthesis of protein, which serves in numerous biological processes, including tissue formation, hormonal regulation, and protein transport. In insects, some amino acids, such as arginine, histidine, leucine, isoleucine, lysine, tryptophan, valine, methionine, and threonine, are classified as essential and must be obtained through dietary sources due to the inability of the insects to synthesize them [107]. Generally, insects possess functional genes for digestive enzymes that enable them to digest proteins directly. Proteolytic enzymes are classified into subcategories based on their specific functions, such as Serine proteases (Trypsin and chymotrypsin), Metalloproteases (Carboxypeptidases), and Cysteine proteases (Cathepsin). These enzymes digest proteins from the substrate into the resulting amino acids which are then absorbed by the gut cells and transported through the haemolymph like blood in vertebrates [108].

In Diptera, protein digesting enzymes have been identified in *H. illucens*, *M. domestica* [109,110]. However, the composition of the gut enzymes responsible for the degradation of a diet is a multifactorial process influenced by the feeding behavior, substrate quality and the microenvironment of the gut. In the gut of insects, proteolysis is usually related to the activities of microbes [111,112]. For example, the proteolytic enzymes produced by the gut microbes have been mentioned to increase the totality of the enzymes (trypsin and peptidase) that digest proteins into amino acids in the gut of the *H. illucens* [78,110,113]. In addition to synthesizing new enzymes involved in protein degradation, bacteria can also play a role in the genetic expression of these enzymes in their host. Recent studies have highlighted that the presence of *Citrobacter amalonaticus* in *H. illucens* promotes the expression of the Hitryp1 gene, which encodes for serine protease and the Himtp1 gene encoding for metalloproteinase whose role is the hydrolysis of amino acids [114]. The ability of microorganisms to influence fly metabolism is not restricted to gene expression alone (Table 2), although not all metabolic pathways are yet clearly explained. In their study, Pei et al. [115] highlighted that the bacterium *Bacillus velezensis* EEAM 10B regulates amino acid synthesis in the black soldier fly, leading to an increase in its protein content. The metabolic link between microorganisms and amino acids does not stop at their synthesis alone. Gut microbes in *Drosophila* play a crucial role in enhancing the absorption and utilization of dietary proteins by optimizing the levels of branched-chain amino acids which serve as essential building blocks for protein production [116].

Although bacteria have been the subject of numerous studies on their potential to increase the quantity of amino acids and proteins in edible insects, we will see that fungi and yeasts are other microorganisms that can be added to the substrate of insects, principally for *H. illucens*. While beer yeast can be added to insect diets to simply increase the protein content [117], and protease-producing fungi such as *Rhizopus oligosporus* have also been used to assist *H. illucens* larvae in digesting their feed [118]. These yeast strains have demonstrated the capacity to produce enzymes that boost protein quantity in the diet by pre-digesting substrate proteins [118]. Another equally relevant example of yeast use in mass insect rearing is *Candida* spp. Adding this yeast to the substrate of *H. illucens* increases its protein synthesis [119].

In the Order Coleoptera, edible insects growing in a substrate like *T. molitor*, *Tribolium castaneum* (Herbst 1797) and *Z. morio*, also contain proteolytic enzymes originating from the insect itself [120,121,122,123]. However, the production of proteolytic enzymes can once again come from microorganisms. To mention just one species of bacteria responsible for the synthesis of these enzymes, *Bacillus subtilis* have been identified in the gut of *T. castaneum* [124]. Various studies have highlighted that the insect microbiome can provide amino acids (tryptophan and cysteine) to their host [66,121]. Some bacteria (*Proteobacteria*, *Bacteroidetes*, and *Actinobacteria*) play a crucial role in synthesizing essential amino acids (histidine, isoleucine, leucine, lysine, methionine, phenylalanine, threonine, tryptophan, valine) in the weevil *Cryptorhynchus lapathi* (L. 1758) (Coleoptera, Curculionidae) before their involvement in protein digestion [125]. In addition, bacteria with the capacity to fix atmospheric nitrogen were found in the digestive system of *T. molitor* [126]. These bacteria enable insects to obtain nitrogen products essential for protein synthesis, particularly when they are deficient in their dietary intake [127].

In Orthoptera, such as *G. sigillatus* and *L. migratoria* we find proteolytic enzymes from the insects [69,128]. Information is rarer in this Order, but we know that *Gryllotalpa africana* (Palisot de Beauvois Palison, 1805) (Orthoptera, Gryllotalpidae) also have bacteria-producing proteolytic enzymes [129]. In the orthopteran *Oedaleus decorus asiaticus* (Bei-Bienko, 1941) (Orthoptera Acrididae), certain bacteria appear to be associated with metabolic pathways that facilitate protein digestion [130]. Finally, it has been suggested that bacteria in the gut of *Diestrammena japanica* Blatchley, 1920 (Orthoptera Rhaphidophoridae) are linked to the synthesis of amino acids that could benefit their host [131].

The diversity of proteolytic enzymes and the range of microorganisms producing them highlight the complexity of these symbiotic relationships and the challenge of determining whether bacterial enzymes act as limiting factors in insect digestion. Microorganisms represent a relevant avenue for maximizing the utilization of food waste and producing insects with a high protein content. However, the activity of enzymes on excessive amounts of protein imposes a metabolic burden, resulting in the production of nitrogenous waste products such as ammonia and uric acid [132]. Further research into the mechanisms governing protein digestion and absorption within this symbiotic relationship may provide insight into understanding the ability of the insect to digest organic waste.

**Table 2 insects-15-00611-t002:** Gut bacteria of mass-reared insect species and their identified function related to protein digestion.

Insect Species	Gut Bacteria	Identified Function Related to Proteins Digestion	Reference
*Gryllotalpa africana*	*Kitasatospora cheerisanensis*	Protease production	[129]
*Hermetia illucens*	*Bacillus velezensis*	Amino acid synthesis	[115]
	*Candida* spp.	Protein synthesis	[119]
	*Citrobacter amalonaticus*	Gene promotion	[114]
	*Rhizopus oligosporus*	Protease production	[118]
*Tribolium castaneum*	*Bacillus subtilis*	Protease production	[124]

### 2.3. Lipid

Lipids are the second most abundant nutritional component in edible insects, after protein [133]. It is not surprising that, even in insects, lipids maintain their primary functions of energy storage and as constituents of cell membranes. However, their roles in diapause, reproduction, cold resistance, and the synthesis of hormones, waxes, and pheromones are equally important [134,135]. The fat body is the principal organ for storing lipids in insects, specifically triacylglycerol (TAG) which constitutes the main lipid form composed of glycerol and fatty acids including saturated fatty acids (SFAs) and unsaturated fatty acids (UFAs) with monounsaturated fatty acids (MUFAs) and polyunsaturated fatty acids (PUFAs) [136]. Fatty acids will determine the nutritional quality of edible insect oil. Other types of lipids present in smaller amounts include sterol (e.g., cholesterol), partial glycerides (e.g., diacylglycerides), phospholipids and wax esters [137]. The specific composition of fatty acids can vary greatly among edible insect species. The fatty acid profile of *H. illucens* is predominantly composed of saturated fatty acids (SFA), primarily due to its high production of lauric acid (C12:0) [138]. In contrast, the fatty acid profiles of edible orthopterans and beetles are dominated by unsaturated fatty acids. For example, *A. domesticus* tends to be rich in polyunsaturated fatty acids (PUFA), mainly linoleic acid (C18:2n6), while *T. molitor* is rich in monounsaturated fatty acids (MUFA), mainly oleic acid (C18:1n9) [139]. Similarly, in Lepidoptera, *B. mori* and *G. mellonella* have higher levels of unsaturated fatty acids (UFA), with around 30% SFA. Silkworms are primarily composed of PUFAs, whereas *G. mellonella* has a profile dominated by MUFAs [140]. However, according to Oonincx et al. [141] and Tzompa-Sosa et al. [142], the amount of lipids and fatty acid composition of insects are also affected by sex, life cycle stage, environmental conditions, and nutrition. Focused on the last, numerous studies reported the influence of diet on the composition of fatty acids [143,144,145,146]. As reported by Chapman [108], acylglycerols, fatty acids, galactolipids, phospholipids, and sterols are the major dietary lipids consumed by insects throughout the diet. Sterol and polyunsaturated fatty acids (PUFAs) are essential for insects as structural components of the cell membrane, as secondary metabolites, or as a starting material for steroid synthesis [108,147,148]. Many fatty acids and phospholipids can be endogenously synthesized from dietary carbohydrates by insects themselves; however, edible insects can bioaccumulate fatty acids from their diet. In this regard, *T. molitor* reared on six different substrates demonstrated an increase in total fat content from 0.46 to 9.34% [143]. In fact, when *T. molitor* larvae were fed with beer yeast, they presented a high percentage of MUFA and a low percentage of PUFA compared to wheat flour bread. In contrast, the highest rate of n-6/n-3 ratio and SFA was presented with oat flour. Fasel et al. [149] and Lawal et al. [150] reported that a high dietary n-3/n-6 PUFA ratio, achieved through flaxseed, increases the proportion of PUFA and the n-3/n-6 ratio of mealworm larvae. After adding brown algae (*Ascophyllum nodosum*) to the feeding media of BSF, the ratio of 18:1n-9 to total fatty acids in larvae increases with the increase in seaweed addition ratio, and the concentration of EPA increased linearly with its concentration in the culture medium, but the retention rate decreased [151]. St-Hilaire et al. [152] improved the ~2% EPA + DHA of total fatty acids in BSF larvae by using fish by-product as a substrate.

To break down these dietary lipids into common end products, insects possess various digestive lipases, mainly TAG lipases and phospholipases, which play crucial roles in the lipolysis process [82,108,153]. These lipases work together in the digestive tract of insects, ensuring the adaptability and flexibility of insects to different types of lipids [154,155]. The source of insect lipase mainly includes two aspects: lipase secreted by insects themselves and lipase produced by microorganisms living in the gut (Table 3). In general, intestinal microorganisms can produce lipase, and Proteobacteria stands out as the most prevalent phylum among the insect’s gut microbiota according to Banerjee et al. [111].

In flies, microorganisms play various roles in lipid metabolism, particularly in lipolysis [156]. Many studies indicated that Proteobacteria, Firmicutes, and Bacteroidetes are the main phyla in the gut of *H. illucens* [157], and feeding high-fat diets increased the abundance of Bacteroidetes. Among them, *Acinetobacter*, as a dominant bacterial genus, could secrete lipase, which breaks down TAG into fatty acids and glycerol for cellular use [158]. The isolation of *Bacillus licheniformis* HI169 and *Stenotrophomonas maltophilia* HI121, two bacteria colonizing the digestive system of BSF, has highlighted the positive involvement of their lipolytic role in larval growth and final weight [159]. Microorganism supplementation has shown that *Staphylococcus aureus*, *Saccharomyces cerevisiae*, and *Rhodopseudomonas palustris* increase BSF lipase activity, and in the case of *S. cerevisiae*, this leads to an increase in the lipid content of BSF [160]. Regarding lipid metabolism in flies, lipolysis is not the only factor influenced by microorganisms. Experiments on axenic Drosophila have highlighted that the common presence of *Acetobacter* and *Lactobacillus* can lower the triglyceride content of their host to a normal level [161]. Mutualism between bacteria appears to be part of the normal digestive functioning in *Drosophila*. Indeed, the presence of *Acetobacter fabarum* and *Lactobacillus brevis* reduces TAG storage compared to *Drosophila* with only an *Acetobacter* population. Metabolites produced by *L. brevis* enable *A. fabarum* to perform gluconeogenesis, thus increasing its population. The various metabolic reactions resulting from these bacterial trophic interactions define the nutrient availability for their host [162]. The case of *Spiroplasma poulsonii* is another instance where a bacterium influences the lipid metabolism of its host, *Drosophila*. This bacterium, which proliferates in the hemolymph, consumes and regulates circulating diglycerides in a specific manner [163]. A final example in flies, where microorganisms influence nutrient availability, is that of conjugated linoleic acids (CLA). In their article, Hoc et al. [138] hypothesized that the presence of CLA in the fatty acid profile of BSF larvae could be linked to microbial biohydrogenation of dietary linoleic acids in the insect’s gut, as is the case in ruminants.

In Orthoptera, the involvement of gut microorganisms in lipid-related metabolic pathways appears to be concentrated in the hindgut [164,165]. These functions are less prevalent than those related to carbohydrate and amino acid metabolism, constituting no more than 3.5% of the total functions observed [130,164]. For example, in *Teleogryllus occipitalis*, the absorption of fatty acids by gut microorganisms accounts for only one percent of the recorded functions [166]. Surprisingly, the diet of Orthoptera does not seem to influence these functions. Indeed, in the carnivorous species *Ocellarnaca emeiensis*, lipid metabolism functions are no more abundant than in herbivorous or omnivorous species [167]. Currently, the role of microorganisms in lipid digestion remains poorly studied in these edible insects, given their predominantly herbivorous or omnivorous diet. Kaufman et al. [165] observed a decrease in the enzymatic activities of C4 and C14 esterases in the midgut of axenic *A. domesticus* individuals, suggesting either a potential role of microorganisms in lipid digestion or a stimulation of the cricket’s enzymatic activities. Conversely, the lipid concentration in the hemolymph of axenic *Schistocerca gregaria* individuals is similar to that of normal individuals, indicating a low contribution of microorganisms to lipid digestion in locust [35].

In edible species of Lepidoptera, the involvement of gut microbiota in lipid digestion and assimilation highlights distinct yet complementary roles. In *G. mellonella*, the long-chain fatty acid content remains similar in the larval gut with or without the presence of intestinal microbiota, suggesting that the host primarily degrades beeswax—its natural diet—into long-chain fatty acids independently of its microbiota. However, the increased activity of metabolic processes and the synthesis of secondary metabolites in the intestinal microbiota suggest that these microorganisms are involved in the degradation process following the production of long-chain fatty acids [168]. This implies that *G. mellonella* utilizes its own enzymatic machinery to break down natural and nutrient-rich lipids, with its microbiota acting as secondary degraders in this process. In contrast, the gut microbiota of *B. mori* plays a more direct role in lipid metabolism. Screening for lipolytic activity on Rhodamine B agar plates identified several lipase-producing bacterial strains, including genera such as *Bacillus*, *Brevibacterium*, and *Corynebacterium*. Lipases produced by these microorganisms break down fats into glycerol and fatty acids, which are subsequently absorbed and synthesized into diglycerides and phospholipids in the midgut epithelial cells [169]. The same method was applied to another silkworm species, *Samia ricini*, resulting in the identification of 28 isolates with lipolytic activity, the majority of which belonged to the *Bacillus* genus (71%) [170]. Additionally, alterations in the gut microbiota structure of *B. mori* due to artificial diet were observed. Concurrently, this diet induced variations in lipase activity, further substantiating the connection between gut microbiota composition and lipid metabolism [171]. In the same vein, Liu et al. [172] shows that the contact with PFAS-type contaminants triggered changes in microbiota, such as the abundance of *Brevibacterium*—one of the lipase-producing bacteria [169]. This was also associated with changes in lipase activity. Once again, microbiota and lipid metabolism appear to be linked for *B. mori*. Thus, different gut microbiota regulates diverse physiological processes, and dysbiosis of their composition may alter the gut functions of silkworms. Therefore, while *G. mellonella*’s microbiota supports secondary lipid degradation, the gut bacteria of *S. ricini* and *B. mori* seem to directly facilitate lipid breakdown and absorption, highlighting the essential role of diet and microbiota composition in lipid metabolism across these insect species.

Contrary to other edible insect orders, the role of gut symbiotic bacteria in lipid metabolism for edible species of Coleoptera remains largely unknown. Surprisingly, the bacterial metabolism of lipids in *T. molitor* is still an unexplored topic (Table 3). A recent study by Mao et al. [173] provided some clarification about the relationship between intestinal microbiota and lipid metabolism in the yellow mealworm. In general, the Actinobacteriota phylum was found to correlate with sphingolipids, the major components of the lipid bilayer in cell membranes, suggesting a potential role in their regulation. Conversely, the bacterium *Weissella* showed a close relationship with cholesterol, indicating a potential inhibitory effect. Among the significantly correlated bacteria, only *Weissella* demonstrated a positive correlation with the synthesis and accumulation of lactosylceramide, a class of sphingolipids involved in the regulation of cell differentiation and proliferation [173]. For non-edible Coleoptera, some studies have been conducted. Jing et al. [125] reported that the gut microbiota of *Cryptorhynchus lapathi* (Coleoptera, Curculionidae) has a relatively small effect on lipid digestion. In contrast, in *Nicrophorus vespilloides* (Coleoptera, *Silphidae*), there is a correlation between the biosynthesis of sterol and fatty acids and the presence of *Yarrowia*, an intestinal symbiont of the species [174]. However, the current literature on the relationship between intestinal microbiota and lipid metabolism is limited for edible species of Coleoptera, highlighting the necessity for additional research to explore this area more comprehensively.

In summary, there are increasing reports on the relationship between insect gut microbiota and insect lipid digestion, with many studies related to the species characterization of gut microbiota that can release lipase (Table 3). Fatty acid composition is a major factor in the nutritional quality of edible insects. Unlike carbohydrates and proteins, fatty acids present in the diet can be directly bioaccumulated by insects such as *H. illucens* and *T. molitor*, thereby enhancing their nutritional value. While the contribution of gut microorganisms to lipid digestion appears minimal, it is crucial to avoid microbial interference with the bioaccumulation of these high-nutritional-value fatty acids, particularly in *H. illucens*, which is naturally rich in SFAs. Additionally, the isomerization of fatty acids by microorganisms remains an unexplored area. This could be a relevant research avenue for the production of high-value molecules, as suggested by Hoc et al. [138] for conjugated linoleic acids.

**Table 3 insects-15-00611-t003:** Gut bacteria of mass-reared insect species and their identified function related to lipids digestion.

Insect Species	Gut Bacteria	Identified Function Related to Lipids Digestion	Reference
*Bombyx mori*	*Bacillus* sp.	Lipase production	[169]
	*Brevibacterium* sp.		
	*Corynebacterium* sp.		
*Hermetia illucens*	*Acinetobacter* sp.	Lipase production	[157]
	*Bacillus licheniformis* HI169	Lipolytic activities	[159]
	*Stenotrophomonas maltophilia* HI121		
	*Staphylococcus aureus*	Lipolytic activities increasing	[160]
	*Rhodopseudomonas palustris*		
*Samia ricini*	*Bacillus* sp.	Lipolytic activities	[170]

### 2.4. Vitamins and Minerals

Vitamins are organic substances essential to the optimal functioning of metabolism, from energy production to the endocrine and nervous systems [175]. Vitamins are classified according to their solubility. Water-soluble vitamins include vitamins C and B vitamins. Fat-soluble vitamins include vitamins A, D, E, and K [175,176]. Insects cannot synthesize all the vitamins their bodies require for proper functioning [38]. Consequently, they must rely on the food they ingest or the microbes inhabiting their digestive systems to obtain the necessary vitamins [177]. Depending on the vitamins, sources can be numerous and varied, ranging from animals to plants [178,179].

The synthesis of various B vitamins by the insect microbiota has been well-studied and has been the subject of literature reviews [180]. The use of probiotics, especially *Lactobacillus* and *Bifidobacterium*, offers a valuable source of B vitamins to insects. These fermentative bacteria produce B vitamins which can be released into the substrate [181,182]. The use of microorganisms to enrich B vitamins is particularly interesting in insects developing in a substrate such as *T. molitor* and *H. illucens*. Specifically, studies have highlighted that supplementing *T. molitor* with probiotics enhanced its nutritional quality by elevating its concentration of vitamin B12 [181,183]. Moreover, research indicates that *Bacillus* and *Gracilibacilus* can improve the conversion rate and growth of *H. illucens* by offering substantial levels of vitamin B2 [115]. Supplementing these insects with B vitamins is particularly interesting from the perspective of animal nutrition. Indeed, vitamin B12 is essential for livestock animals to combat anemia and demyelination of nerve cells [184,185]. Vitamin B2, on the other hand, plays an essential role in energy regulation, redox mechanisms, and in the synthesis of vitamins B6, B9 [186].

Minerals play also various essential roles in the bodies of animals: osmoregulation, thermoregulation, oxygen transport, skeletal mineralization, enzymatic reactions, and the functioning of the nervous and endocrine systems [187,188]. As for vitamins, these elements, not synthesized by insects, come from hydration and nutrition. Insects, having a significantly different skeletal structure and vascular system from mammals, may explain why they have not been the focus of such in-depth studies on the involvement of bacteria in mineral absorption processes [189,190]. There have been studies exploring the involvement of microorganisms in facilitating the detoxification of ions, particularly heavy metals, in insects (Table 4). Examples of endosymbioses where bacteria protect their hosts against iron (Fe) include *Sodalis glossinidius* and the tsetse fly *Glossina* sp. [191], *Spiroplasma poulsonii* and the fruit fly *D. melanogaster* [192], and *Wolbachia* and the wasp *Asobara tabida* (Nees, 1834) [193]. In the case of *M. domestica*, the presence of copper (Cu^2+^) ions alters the bacterial communities inhabiting its gut flora and inhibits larval growth. The bacterium *Pseudomonas aeruginosa* Y12 can utilize this ion, thereby detoxifying the digestive system of the fly larvae and lifting the inhibitory growth effect of Cu^2+^ [194]. Additionally, Yin et al. [195] demonstrated a similar copper detoxification effect by the bacterium *Klebsiella pneumoniae* in the digestive system of *M. domestica*. Concerning edible insects, the presence of copper and cadmium significantly alters the microbial populations in the gut flora of *H. illucens* [196]. BSFL can be used for recycling livestock manure which may contain heavy metals such as copper, zinc, cadmium, chromium, lead, arsenic, and mercury [197,198]. Therefore, exploring microorganisms that may assist *H. illucens* larvae in detoxifying the manure they consume would be highly intriguing.

Insects, much like meat, are an interesting source of minerals and vitamins, particularly cobalamin, also known as vitamin B12 [18,199,200]. However, our understanding of the involvement of insect microbiota in vitamins synthesis or minerals assimilation still has many gaps. These studies offer improved development for insects and enhance their nutritional qualities. Future research on this topic is thus of significant importance to provide new sources of high-quality vitamins and minerals. Interesting avenues of investigation would include studying the impact of microorganisms on mineral absorption or even on the transport proteins of these minerals [201,202].

### 2.5. Contaminants

As previously mentioned, insects have a strong association with microorganisms to increase the degradation of some organic and synthetic molecules [36,203]. Microorganisms release different types of enzymes, such as oxidoreductases, oxygenases, hydrolases, peroxidases, phosphodiesterases, and lipases which contribute to the degradation and transformation of contaminants into substances less toxic and harmful. Based on this, the concept of bioremediation was developed as a biological technique studied and used to remove contaminants from the environment through microorganisms [204,205,206]. Recently, some studies focused on the understanding of this capacity from intestinal microorganisms of insect species in the detoxification and degradation of many contaminants and xenobiotics substances such as pesticides, mycotoxins, antibiotics, and plastics [207,208] (Table 5). As reported by Itoh et al. [209], Blanton and Peterson [210], Jaffar et al. [211], and Siddiqui et al. [212], some symbiotic microbial species isolated from insect gut can contribute to detoxifying pesticides thanks to the potential natural enzymes produced. Singh et al. [213], demonstrated that some bacteria, such as *Pseudomonas* sp. ChlD, *Klebsiella* sp. F-3, *Stenotrophomonas maltophilia* CH-y, *Ochrobactrum intermedium* 13.9, and *Bacillus* sp. C-2 can biodegrade the chlorpyrifos efficiently [213]. Purschke et al. [214] showed that *H. illucens* larvae exposed to high concentrations of pesticides (chlorpyrifos, chlorpyrifos-methyl, and pyrimifos-methyl) did not show pesticide bioaccumulation, and even showed a decrease in pesticide concentration in the residual substrate. This was attributed to the increase in the microorganism diversity and number in the larval gut responsible for pesticide biodegradation. Edible insects can tolerate often high concentrations of mycotoxins thanks to metabolic enzymes such as glycosyltransferase and cytochrome P450 monooxygenase as well as gut bacteria [215,216]. Suo et al. [217] isolated intestinal bacterial strains from *H. illucens* that contributed to Aflatoxin B1 degradation. Finally, antibiotics such as tetracycline, oxytetracycline, ciprofloxacin, tylosin and enrofloxacin found in manure were also found to be degraded by *H. illucens* and *M. domestica* larvae thanks to the intestinal microbiota of both species [218,219,220,221,222]. For example, ciprofloxacin was found to be degraded by *Klebsiella pneumoniae* BSFLG-CIP1 and *Proteus mirabilis* BSFLG-CIP5 from *H. illucens* intestinal microbiota [222].

Among the major pollutants, plastic stands out as one of the most pervasive and concerning for the environment. Global plastic production increased from 234 million tons in 2000 to 367 million tons in 2020, with 55 million tons produced only in Europe, due to its versatile properties [223]. However, significant environmental problems have been reported due to the excessive use (production and consumption) of plastics as they are long-lasting and can remain in the environment for centuries, causing pollution and harm to wildlife and ecosystems [224]. One of the promising suggested solutions to tackle plastic-related environmental challenges is the use of microorganisms for plastic biodegradation [225,226,227,228]. Microorganisms can biodegrade plastics by secreting enzymes that break down polymer chains into smaller molecules, such as multimers and dimers, which are subsequently incorporated into microbial cells and used as the carbon source in the microorganism energy production cycle [226,228]. This process can occur naturally in the environment but can also be induced by adding certain microorganisms (bacteria or fungi) or enzymes to plastic waste [229]. In the last few years, some edible insect species able to ingest synthetic polymers were highlighted by several studies such as *T. molitor*, *Tenebrio obscurus* Fabricius 1792 (Coleoptera Tenebrionidae), *Z. (Z.) atratus*, *A. diaperinus*, *G. mellonella* and *H. illucens* [222,230,231,232]. However, they are generally unable to biodegrade plastic on their own thanks to their interaction with their gut microorganisms [222,230,231,232]. The study of synthetic polymers ingestion by insects focused principally on polyethylene (PE), low-density polyethylene (LDPE), polyethylene terephthalate (PET), polyurethane (PU), polystyrene (PS), polypropylene (PP) and polyvinyl chloride (PVC). The degradation enzymes provided by the gut microorganisms attack the surface of plastic allowing depolymerization into polymeric monomers which are then turned to form fatty acids decomposed by insect biological metabolism [222,233]. Currently, the studies involved in intestinal microbiota analysis are based on 16S rRNA sequencing which is the most popular molecular-based approach to explore associated microbiota in insects that allows the characterization of different microbial taxonomic groups [234]. For example, 56 species intestinal bacteria from 25 different genera have shown the ability to biodegrade PE in *G. mellonella* [235].

One of the most common practices to highlight the role of these gut-related microbes is to add antibiotics with a large spectrum to an insect diet leading to the suppression of the microbe community in their digestive tract. An addition of gentamicin, a bactericidal antibiotic, inhibited the depolymerization of PS by *T. molitor* and *T. obscurus*, and of PP and PVC by *T. molitor* [236,237,238,239,240,241,242]. Similarly, the depolymerization of PE, PS and PP was found to be impossible by individuals of *Z. (Z.) atratus* deprived of their intestinal microbiota by the same antibiotic [238,243]. Nevertheless, it has been reported that the microbiota is not always essential for the depolymerization of the plastic but only enhances it. This is the case of PE biodegradation by *T. molitor*, *T. obscurus* and *G. mellonella* or the beeswax as well as PS biodegradation by *G. mellonella* [235,238,241,244].

Many studies have focused on characterizing the gut microbiota during plastic consumption to highlight a potential change in the community structure of the digestive tract (i.e., the composition and organization of the microorganisms found). The microbiomes’ structure as various clusters associated with the different diets were observed (Table 5), notably between the control diet and the plastic-based diet (PS or PE) in *T. molitor*, *Z. (Z.) atratus*, and *G. mellonella* [245,246,247]. Distinct clusters between the microbiomes of *T. molitor* and *Z. (Z.) atratus* fed with bran and those fed with PU or PP were also discovered [238,248,249,250].

To characterize these changes in structure more accurately, additional measures were also used, such as the specific richness. The quantity of Operational Taxonomic Units (OTUs) found can be an indicator of the diversity of bacterial community. It is frequently combined with other microbial variety indexes like the Shannon or Simpson index that quantify the dominance and evenness of species. However, the findings of various studies on this subject are inconsistent. Some studies have shown a decrease in diversity following PE and PS consumption by *T. molitor* and *Z. (Z.) atratus* or in *G. mellonella* [244,251]. On the other hand, Jiang et al. [245] showed the opposite pattern for the same insects consuming the same plastics. Finally, a third category demonstrates comparable levels of diversity between a plastic-based diet and a conventional diet [247,252,253]. Further studies are, however, needed to extend our knowledge about these effects and the factors that influence them.

The abundance of species found in this community is another important factor to consider. A change in the abundance can indicate preferences of these species towards specific diets and particular OTUs associated with plastic degradation can be potentially identified. For this purpose, relative abundance analysis and differential abundance analysis are commonly used. Relative abundance analysis can be used to characterize the overall composition of the microorganism community, whereas differential abundance analysis can be used to identify species that differ significantly between sample groups. The two methods are complementary and are sometimes used in tandem to obtain a more complete picture of species distribution in a microorganism community. Some studies have been able to reveal specific OTUs strongly associated with plastic-based diets. For instance, *Bacillus aryabhattai*, *B. megaterium* and *Bacillus* sp. were associated with LDPE and PS degradation when ingested by *Z. (Z.) atratus* [246]. Brandon et al. [252] identified two OTUs strongly associated with PE and PS consumption by *T. molitor*: *Kosakonia* sp. and *Citrobacter* sp. *Citrobacter* sp. was also associated with PE consumption in *Z. (Z.) atratus*, while the PU group, which showed the lowest microbiota diversity, presented *Enterococcus* and *Mangrovibacter* as the dominant genera [240,249]. *Enterococcus* sp. and *Spiroplasma* sp. were associated with LPDE consumption in *T. obscurus* and *T. molitor* [238]. *Enterobacter* sp. has also been reported in the gut of *G. mellonella* fed with PS and PP [254]. Several studies have confirmed this increase in Enterobacteriaceae after ingestion of various types of plastics by all the previously mentioned insect species [236,245,248,253]. *Enterococcaceae* and *Streptococcaceae*, mainly of the genus *Lactococcus*, are also reported to be associated with plastic-based diets [250,253].

Other edible insects have been studied to a lesser extent. The capacity of *A. diaperinus* to ingest PS was investigated in a recent study, in which the rRNA based on the gut microbiota revealed different microbial taxa between PS-fed larvae and control ones [255]. Following this study, an enrichment bacterial culture in a liquid carbon-free basal medium with PS film as the sole carbon source, obtained from the intestine of PS-fed larvae of *A. diaperinus* was analyzed [256]. From 16S rRNA gene amplicon sequencing, the predominant taxonomic groups were *Klebsiella*, *Pseudomonas*, and *Stenotrophomonas*, which are among the main microbial taxa involved in PS degradation [257,258]. Even *H. illucens*, which is a widely studied insect for its potential role in organic waste biodegradation and conversion into valuable products such as protein, fats and chitin, is also being studied for its capacity to ingest plastics [259,260]. Despite this, there are still no studies on the intestinal microbiota of *H. illucens* larvae that ingest synthetic polymers; Romano and Fischer (2021) discovered that larvae exposed to PP microplastics showed a high level of propionic and butyric acid, which was not present in the larvae of the control diet [261]. Considering the bacteria origin of short-chain fatty acids (SCFA), this could be attributed to changes in intestinal microbiota caused by the microplastic presence [262]. These results could encourage future research to check the possible biodegradation of plastics by *H. illucens* and to explore new candidates for plastic biodegradation.

It is widely accepted that the insect gut serves as an important habitat for microorganisms and crucial enzymes involved in plastic biodegradation. According to some studies, the ability of insects to break down long-chain polymers in petroleum-based plastics is somewhat suppressed when intestinal microorganisms of the insect are removed by antibiotics [241,243]. This suggests that the intestinal microorganisms of insects play a significant role in the degradation of plastics. For example, a strain of *Pseudomonas aeruginosa* was isolated from the digestive tract of *Z. (Z.) atratus* and showed its ability to degrade PS via an enzyme: serine hydrolase [263,264]. Similarly, the strain of *P. aeruginosa* was isolated from the gut of *Z. (Z.) atratus* fed with a PS diet [263]. The PS-degrading ability of *P. aeruginosa* was examined by measured changes in atomic composition and chemical structural changes using X-ray photoelectron spectroscopy, Fourier-transform-infrared spectroscopy, and nuclear magnetic resonance to confirm the formation of carbonyl groups during PS biodegradation. Another study reported that in the screening process of *Z. (Z.) atratus* larvae fed with Styrofoam developed a new BIT-B35T strain which based on phylogenetics results, genome-relatedness, phenotypic characteristics, and chemotaxonomic analyses belongs to a novel genus in the Enterobacteriaceae family (*Intestinirhabdus alba*) [265]. These insects could constitute an interesting reservoir of microorganisms as well as enzymes capable of depolymerizing plastic. Nevertheless, correct identification of taxa and enzymes involved in the degradation of plastics, changing of polymer structure and discovery of biodegradation products are needed to find a promising strategy for the depolymerization of plastic for recycling or to convert plastic into high-value by-products as biodegradable polymers.

Plants are also a source of contaminants that should not be overlooked. Indeed, secondary metabolites are produced by plants, often in response to herbivore attacks. These molecules are chemically diverse but possess high biological activity and can be toxic to many organisms, including insects [266,267]. Due to the nature of the food given to mass-reared insects, they could ingest these compounds. For example, glycoalkaloids are naturally present in foods containing potato protein and have already been tested as food for cricket species such as *A. domesticus* and *G. bimaculatus* [268]. However, these molecules have shown low bioaccumulation in these insects, and the ingested amounts did not affect their performance [268]. Several defense mechanisms in the insect’s digestive system could explain this low bioaccumulation. First, the insect’s gut can act as a physical barrier, preventing the absorption of certain specific molecules [269]. In *T. molitor*, for instance, it has been demonstrated that some glycoalkaloids can pass through the intestinal barrier and can still be detected throughout the larva’s body 24 h after ingestion [270]. Furthermore, Winkiel et al. [271] observed an effect on lipid metabolism in *T. molitor*, with a decrease in the amount of triglycerides in the fat body and a modulation of the fatty acid profile following the ingestion of these molecules. It is, therefore, very interesting to study the detoxification mechanisms that can reduce the absorption of these secondary metabolites by their degradation through the action of endogenous detoxification enzymes or via the microorganisms present in the insect’s digestive system [269]. Few studies have focused on these molecules related to edible insect species, while many have been conducted on crop pests. For example, the gut microbial community of *T. molitor* has shown a detoxifying effect on secondary metabolites such as glycosides, but the underlying mechanisms have not been explained [98]. Studies on this topic are still underexplored despite the confirmed presence of secondary metabolites in plant by-products and their potential bioaccumulation and toxicity in insects.

**Table 5 insects-15-00611-t005:** Gut bacteria of mass-reared insect species and their identified function related to contaminants digestion.

Insect Species	Gut Bacteria	Identified Function Related to Contaminants Detoxification or the Polymer with Which the Bacteria Species Are Associated	Reference
*Alphitobius diaperinus*	*Klebsiella* sp.	Associated with PS consumption	[256]
	*Pseudomonas* sp.		
	*Stenotrophomonas* sp.		
*Galleria mellonella*	*Enterobacter* sp.	Associated with PP and PS consumption	[254]
*Hermetia illucens*	*Klebsiella pneumoniae* BSFLG-CIP1	Ciprofloxacin degradation	[222]
	*Proteus mirabilis* BSFLG-CIP5		
	*Stenotrophomonas acidaminiphila*	Aflatoxin B1 degradation	[217]
*Tenebrio molitor*	*Kosakonia* sp.	Associated with PE and PS consumption	[252]
	*Citrobacter* sp.	Associated with PE and PS consumption	
	*Spiroplasma* sp.	Associated with LDPE consumption	[238]
	*Enterococcus* sp.	Associated with LDPE consumption	
*Tenebrio obscurus*	*Spiroplasma* sp.	Associated with LDPE consumption	[238]
	*Enterococcus* sp.	Associated with LDPE consumption	
*Zophobas atratus*	*Bacillus aryabhattai*	PS degradation	[246]
	*Bacillus megaterium*		
	*Bacillus* sp.		
	*Pseudomonas aeruginosa*	PS degradation via serine hydrolase	[263]
*Zophobas atratus*	*Citrobacter* sp.	Associated with PE consumption	[249]
	*Enterococcus* sp.	Associated with PE, PS and PU consumption	
	*Dysgonomonas*	Associated with PS consumption	
	*Sphingobacterium*		
	*Mangrovibacter* sp.	Associated with PU consumption	

## 3. Discussion

Mass rearing limits the dietary choices available to insects, often exposing them to feed sources distinct from their natural resources, such as agro-industrial by-products or organic waste. Firstly, the diet may not fulfil all the insect’s nutritional requirements. Secondly, even if the substrate is nutrient rich, the insect’s internal enzymatic processes may be insufficient for the provided diet, leading to limited access to nutrients. The consequences of these mismatches for the insect can include (i) behavioral changes, increased ingestion to offset nutrient deficiencies, (ii) a slowdown or halt in the insect’s development, and (iii) reduced fecundity [272,273,274]. The consequences for rearing are (i) a low bioconversion rate, (ii) alterations in the insect’s nutritional composition, (iii) prolonged production processes, and (iv) an increased production of rearing residues. Currently, the understanding of the insect microbial community’s composition and the beneficial roles of its constituent species is expanding. The wide variety of microorganisms found in edible insects brings with it a significant diversity of metabolic functions. Various bacteria have the capacity to produce proteolytic enzymes in the digestive system of edible insects. For instance, some bacteria can enhance the expression of *H. illucens* genes encoding for proteolytic enzymes [114]. On one hand, some microbes enable edible insects to obtain the proteins they need even when they are growing on a nutritionally poor substrate [115,116]. The metabolic usefulness of bacteria in insect farming goes far beyond proteins; bacteria such as *R. rhodochrous* can increase the quantity of short-chain fatty acids in *H. illucens* while lowering the proportion of monounsaturated fatty acids [28]. The use of probiotics, on the other hand, partially meets the vitamin B needs of insects. Therefore, their use in mass rearing could thus make edible insects an interesting source of vitamin B [181,182]. Another key aspect of the involvement of microbes in the rearing of edible insects is their potential for detoxifying xenobiotics. The insatiable appetite of certain insects, combined with enzymes such as oxidoreductases, oxygenases, hydrolases, peroxidases, phosphoesterases, and lipases, enables them to eliminate various toxins. *H. illucens* larvae do not exhibit bioaccumulation of pesticides or antibiotics but rather a decrease in these substances in the food chain. These phenomena can be attributed to their efficient detoxification system and their intestinal microbiome, which provides them with detoxification enzymes [214,219]. Once again, microbes prove to be valuable allies in mass rearing, ensuring that insects become a safe nutritional source.

In general, no microorganisms in the digestive system of mass-reared insects seem to be indispensable to the animals. The diversity of microbial communities in these insect species reflects a certain flexibility and a predominant influence of transient factors [41]. At the level of the holobiont, microorganisms appear to be a promising area of research for improving insect-food compatibility. This can be achieved through research focusing on the following three objectives: (i) diversifying the holobiont’s enzymatic repertoire by increasing the number of enzymes performing the same function or adding new functions, (ii) producing micronutrients to compensate for nutritional imbalances, and (iii) detoxifying diets concentrated in anti-nutritional factors for the insect. Understanding the involvement of microorganisms in insect digestion needs further development for mass-reared species. The distinction between the roles played by exogenous and endogenous enzymes in digestion remains blurred and challenging. Firstly, identifying the taxa constituting the gut microbial community does not provide a detailed assessment of the services offered by this community to the insect. Comparing the bacterial communities of insects fed sterile diets with different compositions is one way to unveil the roles of microorganisms in digestion. However, comparative studies on macronutrient balance must consider not only the proportions of each macronutrient but also their nature and the insect strain. Indeed, very different results for similar proportions are observed among studies. Specifically, for substrate-grown insects like *H. illucens*, substrate composition plays a decisive role in bacterial community diversity [48]. The technofunctional properties of macronutrients dictate the physical characteristics of the substrate, such as its texture or water retention level. In the case of *H. illucens*, the impact of this texture on the development of maggots and their community of microorganisms is currently underestimated [275]. Secondly, identifying the functions of prokaryotes isolated from the insect’s gut does not fully reflect their complete involvement as some species may have a marginal role in the microbial community. In 2011, Jeon et al. [48] isolated a *Bacillus amyloliquefaciens* sp. from the gut of *H. illucens* and observed its cellulolytic activity. However, this isolation was conducted under aerobic conditions. If the microorganism was indeed present in the intestine, it cannot be asserted that it exerted any significant cellulolytic activity within it. Nonetheless, aerobic microorganisms can colonize the substrate and degrade it before ingestion by the insect. They may also produce enzymes that can be acquired by the insect via substrate ingestion [89]. The identification of the total microbial community and isolation are complementary techniques that should be combined for a detailed analysis of the roles of microorganisms in insect digestion. When used in conjunction, these methods allow a comprehensive understanding of the levers and obstacles involved in establishing an insect line possessing a community of microorganisms adapted to a specific substrate. Levers for modifying the holobiont include seeding the substrate or feed and water with cultivated microorganisms and manipulating the abiotic conditions of rearing. These microorganisms may either originate from isolation within the insect itself or other species. For example, the bacterium *Pantoea agglomerans* strain Sga40 was isolated from the gut of a species of Ethiopian locust and inoculated into the gut of *S. gregaria*. The bacterial strain successfully colonized the digestive system of its new host [276].

The main obstacles to holobiont modification are the physiological conditions of the insect’s digestive system and its basal microbiota. Insect gut microbiota homeostasis is regulated physically by the digestive tract, chemically by the insect immune system (Imd, Duox-ROS), and the JAK/STAT signal transduction pathway, and biologically by the symbiotic community itself [277]. In *H. illucens*, for example, substrate seeding would primarily influence the anterior part of the midgut [31]. Encapsulation techniques for microorganisms of interest could be explored to extend the zone of colonization within the digestive system. Regarding the rearing population, density would also be a crucial parameter to consider, especially with *H. illucens*, where high density increases the temperature of the development tank in mass rearing. Consequently, we can potentially anticipate a selection of anaerobic thermophilic bacterial taxa in the digestive system and aerobic thermophilic taxa in the substrate. As with beneficial associations in the wild, the final step will be the creation of horizontal or vertical transmission routes within the mass rearing system to perpetuate a holobiont lineage. The most effective method for achieving this would be through natural vertical transmission, as, once established, the holobiont would be self-sustaining. However, as the life cycle and habits of mass-reared insects are hardly compatible with the vertical transmission of intestinal bacteria, recovering a fraction of the substrate to inoculate the next-generation substrate could be an interesting approach.

A few studies on the co-conversion of by-products indicate that synergy between insects and microorganisms is possible [278,279]. Firstly, substrate fermentation can aid in conserving surplus substrate, like hay silage in livestock production [278,280]. Secondly, studies show that pre-treatment with microorganisms can enhance substrate conversion by insects. For example, the valorization of soybean curd residue by *H. illucens* was improved with the addition of *Lactobacillus buchneri*, which increased conversion efficiency and enhanced the nutritional value of *H. illucens* larvae [281]. Similar results were observed with dried distillers’ grains with solubles fermented by adding *Lactobacillus plantarum* [282]. In 2024, Alciatore et al. [280] demonstrated that maintaining agricultural by-products rich in lactic acid bacteria in anaerobic conditions for several days achieved the same bioconversion rate with *H. illucens* as using a commercial bacterial inoculum containing *L. plantarum* and *Enterococcus faecium*.

Research has mainly focused on the pretreatment of substrates for *H. illucens*. Interestingly, similar studies for breeding edible Coleoptera are not yet available, despite being common practice among amateur ornamental beetle breeders. Producing large adult specimens often involves feeding larvae with fermented woody substrate, commonly referred to as “Flake Soil” or “Black Soil” on forums, social networks, and commercial sites (author observations). This practice, frequently mentioned for rearing Cetoniidae, Dynastidae, and Lucanidae, could inspire the development of lignocellulosic agricultural waste conversion using edible Coleoptera. Currently, these wastes are still underutilized worldwide [283]. Additionally, they are not yet considered a valuable nutrient source for mass-reared insects due to their high insoluble carbohydrate content.

Palm pest genera like *Oryctes* and *Rhynchophorus* could be promising candidates for managing the lignocellulosic wastes, given their xylophagous diet and the cellulolytic activities of their gut microbiota [100,101]. Since the 2010s, these species have been farmed in the tropics to ensure more consistent food availability for the population and increase farmers’ income sources [284,285]. However, research into optimizing breeding methods and feeding trials remains limited compared to species like *H. illucens* or *T. molitor*. A synergistic avenue of research appears to be emerging due to common areas of interest. Indeed, the development of the mass rearing of xylophagous Coleoptera species could benefit from previous studies conducted by pest management sector on their gut microbiota.

## 4. Conclusions

This review has provided an overview of the current state of knowledge regarding microbes associated with edible insects. Furthermore, it offers insight into areas that still require further research. Identifying the most suitable yeast and bacterial species is crucial for achieving optimal yields in the mass-rearing of edible insects. These microorganisms have been extensively studied in this field and have shown the greatest potential. Most studies have focused on *H. illucens* and *T. molitor*, likely because these are two insect species commonly reared in various substrates from which they obtain benefits thanks to their gut microorganisms’ activity. Further research on the intestinal microbiota of other edible insect species would be valuable in providing a comprehensive understanding of their functionality. One possible approach could be to apply useful microorganisms directly to the breeding substrate to enhance their capabilities towards the target species. Research into insect-related microorganisms should build on this point. As one of the arguments against edible insects is their potential to bioaccumulate heavy metals, research into microorganisms that may prevent such bioaccumulation could be the key to a successful future for insects in the bioindustry. As discussed in this document, microorganisms have great potential in insect mass production. A particular strength of these microorganisms is the role they can play in protein metabolism, whether by aiding protein digestion via enzyme synthesis, or by participating in amino acid and protein synthesis. The production of fatty acids and short-chain fatty acids (SCAFs), vitamin synthesis and notably the various B vitamins offer both practical applications and avenues for further research. Enhancing the intestinal barrier function of edible insects could also enable the utilization of lower-quality or less absorbable food sources. However, there are still certain gaps regarding the involvement of insect microbiota in their metabolism of ions and minerals. Further investigation is also required to gain insights into the connection between the intestinal microbiota of edible insects and their ability to detoxify and degrade contaminants, such as plastic, mycotoxins, and pesticides, present in the diet. This information could also be valuable for developing industrial applications.

## Figures and Tables

**Figure 1 insects-15-00611-f001:**
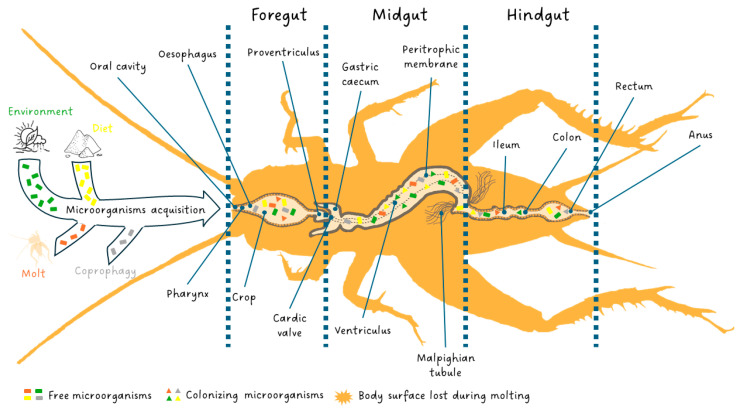
The digestive system of edible insects and their microorganism acquisition through environment, diet, molt and coprophagy. Free microorganisms are symbolized by a rectangle. When ingested by the insect, they may die, circulate in the digestive system and be evacuated in the feces, or colonize the digestive system if the pH and oxygenation conditions are right for them, taking the form of a triangle. Each color designates the origin of the microorganism: green (environment), yellow (diet), dark orange (molt) and gray (coprophagy).

**Table 4 insects-15-00611-t004:** Gut bacteria of mass-reared insect species and their identified function related to vitamins and minerals digestion.

Insect Species	Gut Bacteria	Identified Function Related to Vitamins and Minerals Digestion	Reference
*Hermetia illucens*	*Bacillus velezensis*	Riboflavin synthesis	[115]
*Musca domestica*	*Klebsiella pneumoniae*	Copper detoxification	[195]
	*Pseudomonas aeruginosa*	Copper detoxification	[194]

## Data Availability

The data presented in this study are available on request from the corresponding author.

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
