# Peer review of "Microorganism Contribution to Mass-Reared Edible Insects: Opportunities and Challenges"

_insects, 2024, doi:10.3390/insects15080611_

Round 1
Reviewer 1 Report
Comments and Suggestions for Authors
This is a comprehensive and well-structured review article covering various aspects of edible insect rearing, insect digestive systems, and the role of microorganisms in insect digestion. However, there are several areas that require improvement:
1. The introduction provides a thorough background on edible insect rearing and related research. However, there are some issues:
- On page 2, line 88, "as Hymenoptera and termites" should be "such as Hymenoptera and termites".
- The sentence on lines 95-98 about shedding the exoskeleton may be misleading. Consider rephrasing to more accurately describe how molting affects gut microbiota.
- The legend for Figure 1 (page 3, line 103) needs more detailed explanation, especially regarding microbial acquisition pathways.
- In the sentence spanning lines 184-189, consider organizing the listed bacterial phyla in a specific order (e.g., by abundance or alphabetically) to enhance readability.
- The paragraph starting from line 138 is very long. Consider dividing it into two paragraphs to improve readability.
2. Section 2, "Role of microorganism in the digestive process of natural and synthetic organic," forms the core of the article. While rich in content, the subsections are imbalanced in length. Sections 2.1, 2.3, and 2.5 are notably long, while others (e.g., 2.4) are relatively brief. Consider balancing the content across sections. Specific suggestions:
- In "2.1. Carbohydrates," reorganize the content by types of carbohydrates or insect species to improve structure and readability.
- Similarly, in "2.2. Protein," consider reorganizing to highlight differences in protein digestion capabilities among insect species.
- Sections 2.3 to 2.5 are better organized in comparison.
3. The discussion section effectively summarizes the role of microorganisms in insect digestion and suggests future research directions. This section is well-constructed.
4. The references section includes a wide range of relevant literature, demonstrating the authors' extensive knowledge. However, there are inconsistencies in formatting. Some references include DOIs while others don't, and journal names alternate between full names and abbreviations. Standardize the format throughout.
This is a content-rich and well-structured review article. With appropriate revisions and refinements, it has the potential to become a significant reference in the field of edible insect microbiology research.
Reviewer 2 Report
Comments and Suggestions for Authors
In the present review, the authors investigated and provided new insights on the connection between the intestinal microbiota of edible insects and their ability to detoxify and degrade contaminants present in the diet. This information would be valuable for developing industrial applications.This is a well-written review, which is based upon recent literature of research in this field. Minor changes are need before publication.
1. Please replace ''GRAM'' with ''Gram'' throughout the manuscript.
2. 333-335: In ref. 93, it is stated that '' B. circulans found to utilize all the polysaccharides and have maximum activity of starch degradation in comparison with other bacterial isolates''. Please rephrase for better accuracy. Bacillus circulans is a Gram-positive bacterium, not negative; please correct.
3. Line 691: Please replace ''gentamycin'' with ''gentamicin''
Reviewer 3 Report
Comments and Suggestions for Authors
The manuscript ‘Microorganism contribution to mass reared edible insect: opportunities and challenges’ by Abenaim et al. has been reviewed.
The paper describes the role of microbiota in different insect species, with particular interest in Hermetia illucens, Tenebrio molitor and Acheta domesticus.
The authors focus their efforts on describing microbiota, physiology, and the role of microorganisms in the metabolism of nutrients, micronutrients and contaminants.
The scope of the paper is clear, and the work is of interest. The manuscript is well-written and the bibliography is adequate. I suggest including more Figures and Tables, to attract the interest of the readers (only 1 Figure in the entire review).
My major concerns are related to the limited examples of the specific secondary metabolites and pathways modified by bacteria.
To make the paper more appealing to the readers of the journal I suggest including more case studies on the connection between by-products and microbiota.
In my opinion, the reported review should be accepted for publication after major revisions.
Reviewer 4 Report
Comments and Suggestions for Authors
The manuscript presents an interesting review regarding the relevance of microorganisms in insects' gut and how they promote the metabolization of nutrients and toxic compounds. I have a few suggestions:
- The introduction topic seems to lack the justification for the development of the study. I believe adding information regarding insects' microbiome and its importance in mass rearing could be interesting. Also, regarding the topics numbering, the Introduction should present a topic of its own (example: 1. Introduction, followed by 2. Insect microbiota).
- The manuscript presents interesting information that could be structured as Tables for easier readability. The text density without tables to synthesize the information could hinder its comprehension.
- I believe the discussion topic is not truly necessary for a review manuscript. It should be renamed and verify if information already presented before is not repeated here.
Round 2
Reviewer 3 Report
Comments and Suggestions for Authors
The paper should be accepted as it is. Thanks for your efforts